# Novel method to measure temporal windows based on eye movements during viewing of the Necker cube

Patrik Polgári[1,2], Jean-Baptiste Causin[1,2,3], Luisa Weiner[1,2,3], Gilles Bertschy[1,2,3], Anne Giersch[1,2,3]*

**1** INSERM U1114, Strasbourg, France, **2** University of Strasbourg, Strasbourg, France, **3** Psychiatry Department, University Hospital of Strasbourg, Strasbourg, France

\* giersch@unistra.fr

**Data Availability Statement:** All relevant data are within the paper and its Supporting Information files.

## Abstract

Bistable stimuli can give rise to two different interpretations between which our perception will alternate. Recent results showed a strong coupling between eye movements and reports of perceptual alternations with motion stimuli, which provides useful tools to objectively assess perceptual alternations. However, motion might entrain eye movements, and here we check with a static picture, the Necker cube, whether eye movements and perceptual reports (manual responses) reveal similar or different alternation rates, and similar or different sensitivity to attention manipulations. Using a cluster analysis, ocular temporal windows were defined based on the dynamics of ocular fixations during viewing of the Necker cube and compared to temporal windows extracted from manual responses. Ocular temporal windows were measured also with a control condition, where the physical stimulus presented to viewers alternated between two non-ambiguous versions of the Necker cube. Attention was manipulated by asking subjects to either report spontaneous alternations, focus on one percept, or switch as fast as possible between percepts. The validity of the ocular temporal windows was confirmed by the correspondence between ocular fixations when the physical stimulus changed and when the bistable Necker cube was presented. Ocular movements defined smaller time windows than time windows extracted from manual responses. The number of manual and ocular windows both increased between the spontaneous condition and the switch condition. However, only manual, and not ocular windows, increased in duration in the focus condition. Manual responses involve decisional mechanisms, and they may be decoupled from automatic oscillations between the two percepts, as suggested by the fact that both the number and duration of ocular windows remained stable between the spontaneous and focus conditions. In all, the recording of eye movements provides an objective measure of time windows, and reveals faster perceptual alternations with the Necker cube and less sensitivity to attention manipulations than manual responses.

**Funding:** The first author (PP) was financed by the French National Research Agency (ANR AutoTime) (https://anr.fr/) and the Grand-Est Region (https://www.grandest.fr/). The funding of the research was provided by a grant attributed to GB (PRI 5582) by the University Hospital of Strasbourg (http://www.chru-strasbourg.fr/). The funders had no role in study design, data collection and analysis, decision to publish, or preparation of the manuscript.

**Competing interests:** The authors have declared that no competing interests exist.

## Introduction

Our perception of the outer world is usually stable: objects and persons do not change abruptly from one moment to another. Yet, when we look at a bistable figure, such as the Necker cube (Fig 1), our perception changes after a while and we perceive the figure differently despite the unchanged sensory input. The Necker cube can be perceived in two different ways and the two interpretations of the picture alternate, each one dominating our consciousness for a few seconds before giving way to the other percept [1]. Perceptual bistability has become a widely used experimental tool to study the mechanisms associated to consciousness, and especially the dynamics of mental and neural activity linked to conscious perception [2,3]. One limit is that alternations rely on the subjects' explicit reports, which include decision criteria and self-monitoring, and can bias results [4–6]. This becomes critical when exploring alternations in clinical populations [7], especially as in those populations, experimenters often privilege short-duration evaluations, due to the fatigability of the patients [8,9]. It has recently been suggested that it may be possible to measure alternations by means of eye movements and fMRI, which were found to be strongly coupled to explicit reports [10,11]. However, these studies relied on moving plaids or gratings and the recording of optokinetic nystagmus. Hence, it can be questioned whether ocular movements are entrained by the motion specifically. As a matter of fact, the tight coupling between perception and eye movements has been recently questioned [12], and dissociations between eye movements and conscious reports have been described even with moving stimuli [13]. Interestingly, dissociations have been reported mainly when the task requires continuous evaluation of the stimuli rather than alternative force-choice tasks [12]. Here we use the bistable and static Necker cube to study the temporal characteristics of alternations over a short, continuous period. We check to which extent changes in ocular fixations define temporal windows that are similar, or not, to the subjective alternations reported by the subjects.

### Why measure time windows?

The definition of time windows is a critical issue to understand how conscious perception is structured in time. Despite the fact that time seems to flow continuously, we have a sense of present moment, corresponding to our subjective experience of being here and now. It has been proposed that the present moment, or 'subjective present', corresponds to the time required to accomplish a mental act in perception, cognition or action [14–16]. The 'subjective present' has no fixed period, but its duration would be between a few hundreds of milliseconds and a few seconds. The Necker cube has been proposed as an operationalization of the concept of the 'subjective present', each percept corresponding to a moment. As a matter of fact, the mean duration of each percept range between 2.0 and 3.2 seconds [17,18]. It has been used in the context of bipolar disorder to evaluate the possibility that there is either a slowing down or

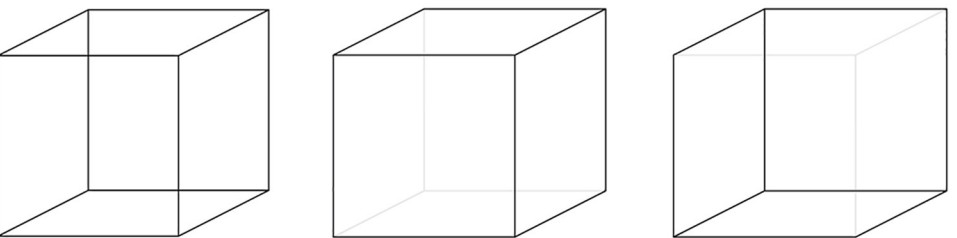

**Fig 1. The Necker cube, an example of ambiguous figure (left), and its two non-ambiguous versions (middle, right).**

an acceleration of thought, depending on the state of the patients [8,9,19]. The dynamics of perceptual alternations during viewing of the Necker cube have also a neurobiological counterpart, since they have been found to be correlated with endogenous brain dynamics [3]. However, whether or not perceptual alternations actually reflect a rhythmicity of our perception has been questioned [20]. Moreover there is more to the temporal structure of consciousness than only the sense of subjective present. Pöppel has proposed that temporal windows of different lengths are embedded in one another, leading to a hierarchical organization of mechanisms characterized by different rhythmicities [14]. When we perceive and interact with our environment, essentially we integrate information at different time scales into a coherent whole, and measuring time windows solely with subjective reports might be misleading. It may thus be useful to collect additional responses to subjective reports.

## Why track eye movements?

Experiments studying spontaneous perceptual reversals of bistable figures are often conducted with the figure remaining on the screen for a certain period of time (for the sake of simplicity from now on we will take the example of the Necker cube) (but see e.g. [1] for a two-alternative forced choice procedure). The participant is instructed to look continuously at the figure and to press a response button each time his/her perception of the Necker cube changes. Hence these measures reflect the viewer's subjective perception, as approximated by his/her explicit manual responses. We call this response explicit, because subjects are explicitly instructed to give a manual response each time they detect a perceptual change. This means that after a perceptual reversal has taken place, the viewer has to make a conscious decision of pressing a button. This subjective response, like any manual response to a subjective phenomenon, includes a response bias that is entirely dependent on the viewer. A growing body of research has shown that the rate of perceptual reversals and reversal times of ambiguous figures are associated to important interindividual differences in healthy individuals [21–24]. This variability might index either the individual temporal characteristics of the perceptual alternation itself, or the individual biases related to the need to give an explicit, subjective response. Here we develop an additional, more implicit measure by means of ocular movements. We call this measure implicit because the instructions do not require the subjects to make any specific eye movements during the task. The subjects are only asked to give their manual responses as reliably as possible while looking passively at the figure. In the present work, the main aim is to have an additional measure of time windows to test whether all measures lead to the same result or not.

## Eye tracking and perceptual bistability in the literature

Some studies combining eye tracking measurements and perceptual bistability have been conducted with static stimuli, but these focused on short periods around the moment of perceptual reversals in order to determine whether ocular movements cause perceptual reversals or vice versa [25–27]. It has been shown that during perceptual reversals the position of fixations are different for the two percepts [26] and fixation coordinates have extreme values at the moment of the reversals [28]. A few other studies have also queried whether it is possible to entrain perceptual reversals via controlled ocular oscillations [29]. To the best of our knowledge, however, research has been mainly conducted on the periods around the perceptual alternations, and it has not been checked whether for static stimuli ocular movements can be used to measure a spontaneous oscillation akin to the manual windows. Hancock et al. (2012) [30] used binocular rivalry, which also leads to perceptual alternations, this time between the information conveyed by one or both eyes. Hancock et al. (2012) found a positive correlation

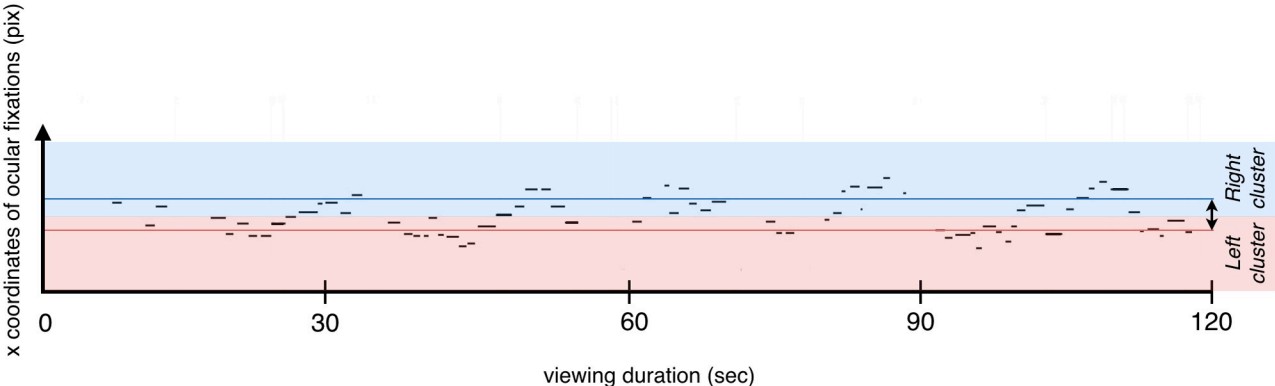

**Fig 2. Illustration of the oscillatory pattern of ocular fixations during viewing of the Necker cube.** The graph represents the x coordinates of ocular fixations as a function of time for one individual participant during the spontaneous condition. The horizontal strips correspond to ocular fixations. The length of each strip is proportional to the corresponding fixation's duration. The closer the value of an x coordinate is to zero, the closer the fixation is to the left side of the screen. The blue shaded part (above) corresponds to the right cluster, whereas the red one corresponds to the left cluster. The blue and red lines correspond respectively to the median of the x coordinates of the right and left clusters. The difference between these medians represents the distance between the clusters (the black arrow on the right part of the graph).

between the rate of perceptual reversals and the rate of saccadic eye movements, pointing toward shared underlying mechanisms. The absolute rate of the two measures differed, however, since saccadic eye movements occurred more frequently than perceptual reversals. It is possible that successive saccades relate to the same subjective perception, but this possibility cannot be checked in the results of Hancock et al. (2012). By using a relatively large picture of the Necker cube and a clustering method to classify saccades, we aimed to measure the frequency of ocular-related time windows relative to the frequency of manual-related time windows. Our approach is directly inspired by the temporal windows model [14]. We noticed an oscillatory pattern in the positions of eye movements between the left and right part of the figure (Fig 2), and we defined temporal windows based on the dynamics of ocular fixations during viewing of the Necker cube. These calculations define 'ocular temporal windows' which are compared to temporal windows based on manual responses. First we checked the validity of the ocular temporal windows in a control condition, during which the physical stimulus presented to the viewer oscillated between two non-ambiguous versions of the Necker cube (Fig 1). In addition, we checked whether ocular and manual time windows have the same duration and vary in the same way in different experimental conditions. After a first condition where subjects were asked to report spontaneous reversals (spontaneous condition), two conditions were used to test the impact of attentional control. In one condition subjects were asked to focus on the same preferred percept as long as possible (focus condition), whereas in the other they were asked to switch as fast as possible between the two percepts (switch condition). It has already been shown in the literature that these instructions yield significant changes in manual windows [25,31,32]. If ocular and manual measures index the same phenomena, then they should be similarly sensitive to attention manipulations, whereas if they reveal different types of windows, they should be affected differently by attention manipulations.

## Materials and methods

### Participants

Twelve subjects (mean age ± SD: 30.8±7.6; 9 females and 3 males) with normal or corrected-to-normal vision participated in the experiment. The project was approved by the local ethics

committee (People Protection Committee "Est-IV"). All subjects gave their informed written consent in accordance with the declaration of Helsinki.

Exclusion criteria included a history of substance abuse, and a history of neurological or psychiatric disorders.

One subject had to be excluded from the analysis of the ocular clusters' spatial coordinates due to technical problems during data acquisition in the control condition. Data presented in the corresponding section are thus averaged over 11 subjects. The rest of the results remained identical whether or not this subject's data were taken into account in the analyses.

## Equipment and stimuli

The experiment was conducted in a quiet room with reduced illumination. Stimuli were generated by a Hewlett-Packard Compaq 8100 Elite 2 computer using programs written on MATLAB software (2007) by MathWorks and PsychToolbox [33]. The visual stimuli were generated on a 21" Sony Triton CRT screen.

During each experimental trial the Necker cube was presented on the screen for 60 seconds. Each side had a length of 12˚ of visual angle, consisting of black lines (0.008 cd/m$^2$ and 0.18˚ thick) on a white background (41.5 cd/m$^2$).

## Eye tracking

Right eye movements were measured continuously throughout the experiment using an infrared video-based eye tracking system (EyeLink CL 1000, SR Research) with a sampling frequency of 1000 Hz, and a spatial resolution of 1024 x 768 pixels. Before each experimental condition, the eye tracker was calibrated by asking participants to repeatedly fixate a 9-point grid. Participants' heads were stabilized using a chin-rest at a distance of 70 cm from the computer screen in order to minimize head movements and errors of measurement.

We registered and analyzed the number of ocular fixations and the mean duration of each fixation. Fixations were defined by stable eye position coordinates for at least 90 ms, and they were excluded if their mean duration was longer than 1000 ms. This eye tracking data processing was carried out using programs written on MATLAB software and Statistica 13.0 software.

## Experimental task and procedure

The experiment consisted of four conditions: spontaneous, focus, switch, and control conditions. In the first, spontaneous condition participants were instructed to manually report the perceptual reversals of the cube that occurred spontaneously, by pressing one of two buttons on a keyboard, each corresponding to a perceived orientation of the cube (left response button for the downward left-facing orientation, right response button for the upward right-facing orientation).

In the focus condition participants were asked to focus on and mentally hold their preferred orientation of the cube and go back to this orientation as quickly as possible in case of reversal. In the *switch* condition participants were instructed to switch as quickly as possible between the two orientations. Just as in the spontaneous condition, participants had to report the perceptual reversals of the cube with button presses. The order of the focus and switch conditions was randomized between subjects, but the experiment was designed so that the spontaneous condition always came first, thus minimizing the effect of voluntary control on perceptual reversals in this condition.

A final control condition was designed to make sure that the subjects were able to detect and reliably report the perceptual reversals. To that aim two modified, non-ambiguous versions of the Necker cube (Fig 1) were presented alternately at the same location, each figure

presented for a duration of 3 seconds. Participants were asked to press a response button each time they perceived an alternation and to choose the key corresponding to the current orientation. The response time recorded in this condition was used to estimate time between the perceptual reversals and the subject's reaction. The key press is necessarily delayed relative to the occurrence of the perceptual reversal, and the reaction time in the control condition was used in the other three conditions, in which there was no physical reversal, to evaluate the time of occurrence of the perceptual reversals.

## Behavioral and eye tracking data processing and analysis

For each subject the preferred and non-preferred percepts of the cube were identified based on the median duration of the reversal times, i.e. intervals of transiently stable percepts based on the subjects' manual responses. It has been shown elsewhere [7,27] that everyone has a preferred percept of the Necker cube, that is one orientation (bottom-left or top-right) is perceived for longer periods than the other, despite the continuous alternation of the two. We extracted the total number of perceptual reversals and the median duration of the reversal times (referred to hereafter as "manual windows"). Preferred and non-preferred percepts were respectively defined by the orientation corresponding to the longer and shorter manual windows.

In parallel we identified ocular fixations, and we quantified them for each subject. Fixations were analyzed using a cluster analysis. The graphic representation of the changes in x and y fixation coordinates over time indeed suggested an oscillation between two positions (Fig 2). Y coordinates appeared to be redundant with x coordinates, hence we focused on ocular displacements on the horizontal axis. It is to be noted that an analysis performed on y coordinates led to similar but less clear results, probably because the two perceptual interpretations of the Necker cube differ more in their horizontal orientation (see similar observations in [26]). Considering the two alternating mental representations of the Necker cube we decided to separate the fixations into two groups, so called "ocular clusters", based on their spatial coordinates. To this aim we used the Expectation-Maximization (EM) algorithm [34] which permits to find for any data the maximum likelihood of belonging to a previously specified number of clusters. Each fixation was classified into one of two clusters (left or right) and each cluster was composed of one or several successive fixations. "Ocular windows" were defined as temporal windows corresponding to the summed duration of ocular fixations inside each cluster. We extracted the total number and the median duration of ocular windows.

In order to confirm the validity of our cluster analysis, we compared ocular windows in the control and in the three main conditions. To this aim, we analyzed the spatial coordinates of fixation clusters in all four conditions (spontaneous, focus, switch, and control). In the control condition we separated the fixation clusters corresponding to the two non-ambiguous versions of the Necker cube. We calculated the median of the x coordinates of fixation clusters corresponding to each non-ambiguous version of the cube. In the other three conditions, we calculated the median x coordinates of the left and right fixation clusters, and the spatial distance between the left and right fixation clusters (median x coordinate for the right cluster–median x coordinate for the left cluster, see Fig 2). The aim was to check whether the x coordinates of the left and right clusters corresponded to the coordinates observed in the control condition.

## Statistical analysis

Data analyses consisted of repeated measures ANOVA and were performed using Statistica 13.0 software by Statsoft©. We added the partial eta-squared ($\eta^2$) as a measure of effect size. Tukey HSD post hoc analyses were used to localize the differences. Correlations were

conducted by computing Pearson's correlation coefficients. The significance level throughout the analyses was set at $\alpha = 0.05$.

We checked the effect of order between the *focus* and *switch* conditions in separate analyses, and found no significant effect. The results presented hereafter are thus averaged over the order between the *focus* and *switch* conditions.

## Results

### Spatial coordinates of ocular clusters

In the first analysis we distinguished the fixation clusters based on their left/right location on the screen with the aim to see if a correspondence in the x axis coordinates of ocular clusters could be found between the control condition and the other three main conditions. We conducted a repeated-measure ANOVA on the coordinates of the ocular clusters with the abscissa of the clusters (left or right) and condition (spontaneous, focus, switch and control) as within-group variables. Left and right clusters had statistically different x coordinates [$F(1, 10) = 112.63$, $p<0.001$, partial $\eta^2 = 0.92$] (with x(left) = 612 vs x(right) = 684 in pixels). We found an interaction between cluster location and condition [$F(3, 30) = 3.84$, $p<0.05$, partial $\eta^2 = 0.23$]. Decomposing the interaction using HSD Tukey's post hoc analysis showed that the coordinates of left and right clusters were separated by a significant distance in all four conditions. Coordinate values are detailed in Table 1.

To understand the origin of the interaction, we calculated the distance between the right and left clusters in each condition (median x coordinates for the right cluster–median x coordinates for the left cluster) (Table 1, also illustrated in Fig 2). This analysis revealed that the distance varied over the four conditions [$F(3, 30) = 3.84$, $p<0.05$, partial $\eta^2 = 0.28$]. Tukey's post hoc analysis showed that the distance between the left and right clusters was larger in the switch condition (108) than in the spontaneous condition (50, $p<0.05$). No other significant difference was revealed between the conditions.

### Number of manual and ocular windows

In the following analyses fixation clusters were distinguished based on their correspondence to the preferred and non-preferred percepts of the Necker cube and were used to define ocular windows.

A one-way ANOVA conducted on the number of manual responses (i.e. button presses reflecting perceptual switches), with experimental condition as a within-group variable, revealed an effect of the condition (spontaneous vs. focus vs. switch) [$F(2, 22) = 6.48$, $p<0.01$, partial $\eta^2 = 0.37$] (Fig 3). Tukey's post hoc test showed that the number of manual responses

**Table 1. Coordinates of left and right fixation clusters in the four experimental conditions.**

| Condition | Left coordinate | Right coordinate | p-value | Right–Left coordinates |
|---|---|---|---|---|
| Spontaneous | 610 | 660 | **<0.05** | 50 |
| Focus | 643 | 703 | **<0.005** | 60 |
| Switch | 599 | 707 | **<0.001** | 108 |
| Control | 596 | 665 | **<0.001** | 69 |

The first two columns in the table show the left and right x coordinates of the fixation clusters in pixels (x = 0 pix corresponds to the left border of the screen), and the third column the p value corresponding to the comparison of the left and right coordinates for each condition. The column on the right shows values of the distances between left and right fixation clusters in the four conditions.

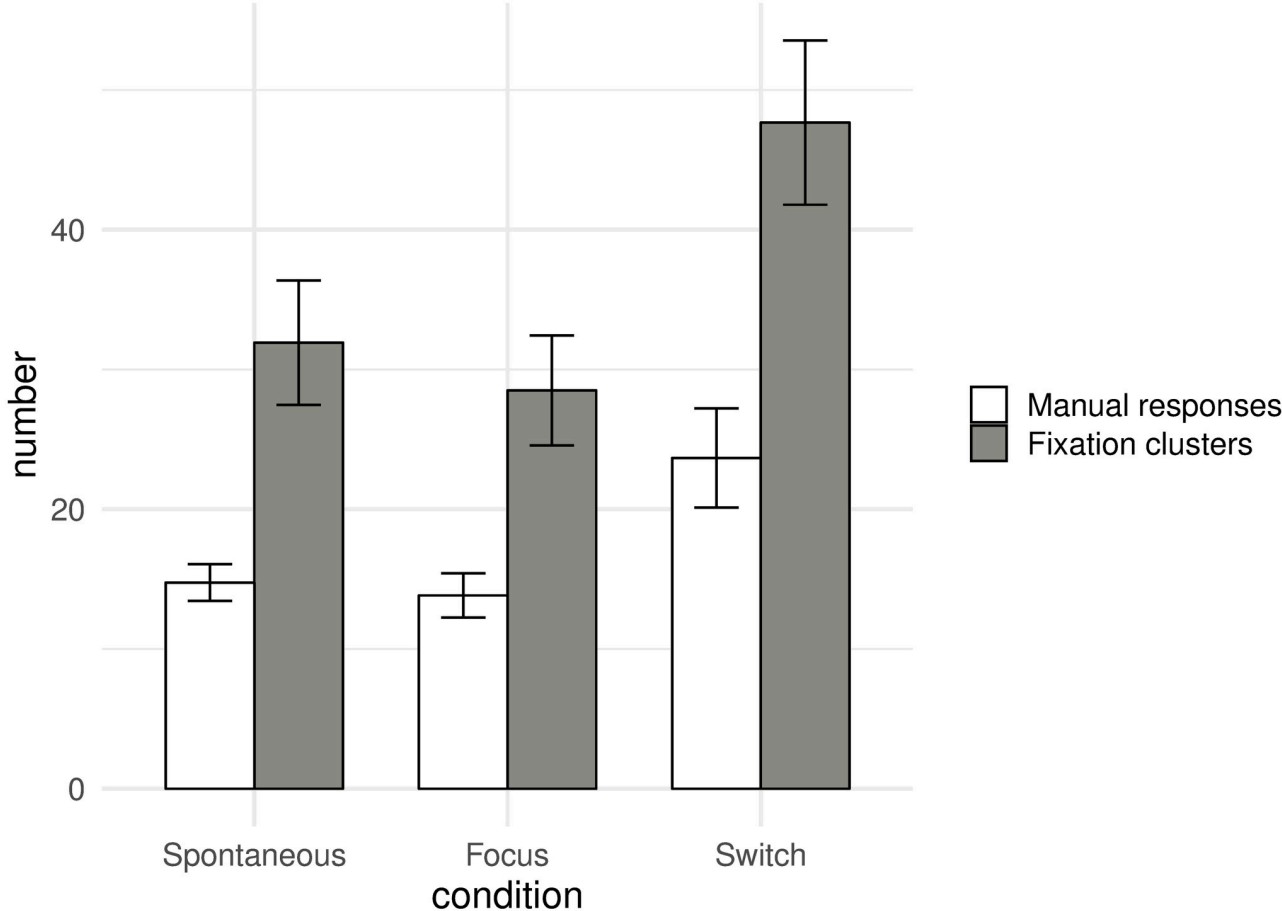

**Fig 3. Illustration of the results for the number of manual responses and fixation clusters.** Number of manual responses (white) and ocular fixation clusters (grey), as a function of the experimental conditions (spontaneous, focus, switch). Error bars represent ± SEM.

increased in the switch condition (23.67) compared to the spontaneous (14.75, p<0.05) and focus (13.83, p<0.01) conditions.

A similar analysis revealed an effect of the condition also on the number of fixation clusters [F(2, 22) = 9.78, p<0.001, partial $\eta^2$ = 0.47] (Fig 3). Tukey's post hoc analysis showed that the number of ocular clusters, similar to the number of button presses, increased in the switch condition (47.67) compared to the spontaneous (31.92, p<0.01) and focus (28.50, p<0.005) conditions.

We conducted an additional analysis to compare the number of manual and ocular windows, with experimental condition and window type (manual vs. ocular) as within-group variables. The results showed that the number of ocular windows was significantly higher than the number of manual windows [F(1, 11) = 18.59, p<0.005, partial $\eta^2$ = 0.63] (36.03 vs. 17.42 respectively). The analysis also revealed an effect of the condition [F(2, 22) = 18.72, p<0.001, partial $\eta^2$ = 0.63]. No interaction was found with the window type.

### Durations of manual and ocular windows

A repeated measures ANOVA was conducted on the median duration of manual windows, with condition (spontaneous vs. focus vs. switch) and preference (preferred vs. non-preferred percepts) as within-group variables. An effect of preference was revealed [F(1, 11) = 36.26,

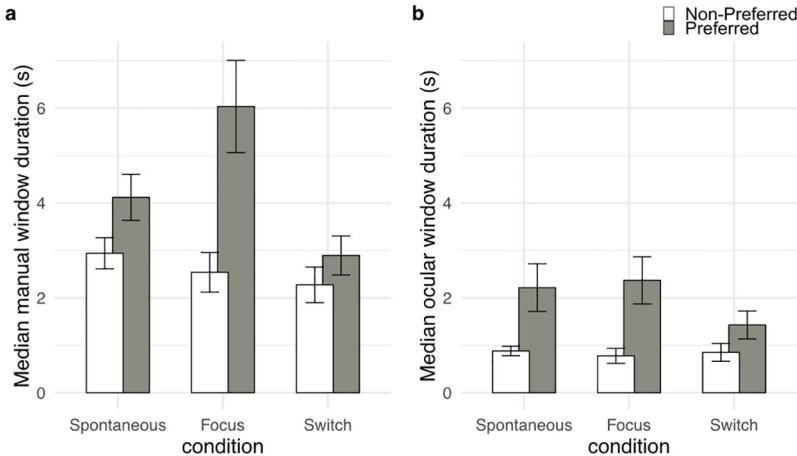

**Fig 4. Illustration of the results for the median duration of manual and ocular windows.** Median duration of preferred and non-preferred (in grey and white respectively) manual (a) and ocular windows (b) as a function of the experimental conditions (spontaneous, focus, switch). Error bars represent ± SEM.

p<0.001, partial $\eta^2$ = 0.77] with a longer median window duration for the preferred percept compared to the non-preferred percept (4.35 s vs 2.59 s, p<0.001). This effect is trivial since preference was defined by the length of the temporal windows. The critical result concerned the impact of the condition (spontaneous vs. focus vs. switch) on the windows' length. We found an interaction between condition and preference [F(2, 22) = 8.65, p<0.005, partial $\eta^2$ = 0.44]. Decomposing the interaction by means of Tukey's post hoc analysis showed that the median manual window duration for the preferred percept in the focus condition (6.04 s) was longer than all the other durations (preferred/spontaneous 4.12 s, p<0.05; non-preferred/spontaneous 2.94 s, p<0.001; non-preferred/focus 2.54 s, p<0.001; preferred/switch 2.89 s, p<0.001; non-preferred/switch 2.28 s, p<0.001) (Fig 4a).

A similar analysis of the median duration of the ocular windows, with condition and preference as within group variables, revealed a main effect of preference [F(1, 11) = 21.89, p<0.001, partial $\eta^2$ = 0.67] with a longer median ocular window duration corresponding to the preferred percept (2.00 s) compared to the non-preferred percept (0.84 s) (Fig 4b). There was no significant effect of condition or interaction between the two factors.

We compared the median duration of manual and ocular windows, with window type (manual vs. ocular), preference and experimental condition as within-group variables. The results showed that ocular windows had a shorter median duration than manual windows [F(1, 11) = 37.01, p<0.001, partial $\eta^2$ = 0.77] (1.42 vs. 3.47 respectively). No interaction was revealed between the window type and the other factors.

## Correlations

No correlation was found between ocular and manual windows.

## Discussion

The main results are, firstly, the correspondence between the ocular fixation clusters when the Necker cube is ambiguous and when the cube changes physically in the control condition. Second, ocular fixation clusters alternate much more frequently than manual responses. Nevertheless the number of temporal windows changes in the same way between the spontaneous and switch conditions, whether measured with eye fixations or manual responses. In contrast,

in the focus condition the durations of manual windows change without being accompanied by a corresponding change in the durations of ocular windows, indicating that ocular and manual time windows are not always sensitive in the same way to attention manipulations.

The difference between ocular and manual time windows can hardly be attributed to the way we calculated ocular windows. Our choice of the "EM" cluster analysis on ocular fixation measurements was validated by the findings on the ocular fixation clusters' spatial coordinates. We found similar coordinates for fixation clusters in the spontaneous, focus and control conditions. This implies that the fixation clusters are located in the same locations on the screen when the two versions of the cube are physically distinct (control condition) and in the spontaneous condition, where the Necker cube stays physically the same and only the subjective perception changes over time. The spatial correspondence between ocular fixations in the two conditions suggests that the spatial locations of the fixations in the spontaneous condition are linked to changes in perception. The correspondence between ocular fixations and percepts is also supported by the fact that manual and ocular windows that correspond to the preferred version of the Necker cube both had longer durations than those corresponding to the non-preferred version of the cube. These results validate the use of the spatial coordinates of fixation clusters to study the dynamics of ocular windows and compare them with that of manual windows.

It can thus be affirmed that ocular clusters alternate much more frequently than the manual windows. This result suggests that the percept does not change each time eyes change position. It may be possible that changes in ocular positions correspond to oscillations between distinct percepts that do not reach consciousness. Such an interpretation would be consistent with observations that eye movements can respond to invisible stimuli [35], i.e. a dissociation between eye movements and perceptual awareness [12]. Regarding studies on bistable stimuli, our results are consistent with those reported by Brascamp et al. (2015) [5], or Kornmeier et al. (2019) [36] who showed in fMRI or EEG evidence of alternations that occur faster than the behavioral responses. As suggested by these authors sensory alternations may occur without reaching consciousness. The possibility that changes in ocular positions reflect an unconscious oscillation between percepts is plausible in the light of models by Logothetis (1998) [37] and Grossberg et al. (2008) [38] which posit that different interpretations for visual information compete for consciousness. Several interpretations of the information would be made available by automatic and unconscious perceptual mechanisms, leading to relatively fast alternations on a non-conscious level, as suggested by ocular movements.

Additional volitional and attentional mechanisms may be needed to decide about the percept and give a manual response. This may contribute to some of the dissociations between ocular and manual windows. For example, the duration of the manual windows that correspond to the preferred percept of the Necker cube increased in the focus condition, suggesting that subjects succeeded in maintaining their preferred orientation of the Necker cube for longer periods. However, neither the number nor the duration of the ocular windows corresponding to the preferred percept in the focus condition were significantly different from the results in the spontaneous condition. This means that the increase in size of the preferred manual windows was not associated with a corresponding ocular change. These results suggest a decoupling between the manual and ocular windows in the focus condition. The literature suggests that a range of factors influence alternations and perception in general [39]. For example, it has been demonstrated that top-down factors, or volitional control [7,25,27,31,32,40,41] can modify the rate of perceptual alternations, as also observed in our focus and switch conditions. As repeatedly observed in focus conditions [42], it is however impossible to completely suppress alternations, confirming the conjoint influences of sensory competition and volitional control. In our study one possible explanation may thus be that the ocular temporal windows

reflect an irrepressible sensory alternation, whereas manual responses would be more susceptible to volitional control. The latter may be used to refrain from perceptual alternations, by vetoing sensory influences, or may be necessary to transform a sensory alternation into a conscious one. Both explanations would account for the ocular fixation clusters' lack of sensitivity to the focus condition, contrasting with the increase in length for the manual time windows.

Contrary to the focus condition, in the switch condition both ocular and manual windows change in parallel. The number of manual responses (i.e. number of perceptual reversals) and the number of fixation clusters both increased in the switch condition in comparison to the spontaneous and the focus conditions. These results suggest that when subjects try to speed up the perceptual reversals of the Necker cube, they do succeed, and this is accompanied by an increased number of ocular movements. In the switch condition changes in ocular positions, i.e. voluntary eye movements may be used to increase the rate of perceptual alternations. This possibility is supported by the analysis of the distances between the left and right cluster coordinates. This analysis revealed that subjects tend to fixate two locations on the screen that are more distant in the switch condition than in the spontaneous condition. We can speculate that this finding reflects the subjects' effort to voluntarily alternate between the two percepts. Caution is required when interpreting these results however, since the duration of ocular windows remains smaller than that of manual windows even in the switch condition. Even in this condition, there is no perfect match between eye movements and manual responses.

In conclusion, we propose a new way of measuring temporal windows with the clustering of ocular fixations. We do not claim that eye movements analyzed the way we did could substitute explicit reports, but instead we believe that eye movements could serve as a complementary measure to explicit reports, inasmuch our ocular temporal windows (ocular clusters) seem to reveal dynamic perceptual mechanisms at a different, more automatic level than the one of conscious perception and perceptual alternations. The ocular temporal window measure is validated by the similarity of the coordinates of the fixation clusters for ambiguous and non-ambiguous versions of the Necker cube, and by the parallelism of the effects of preference and switch on ocular and manual windows. The ocular windows suggest that there is a spontaneous rhythm at an implicit level, which might reflect perceptual alternations between possible interpretations of sensory information. These alternations are faster than alternations at the conscious level, confirming that alternations occur at different time scales. It remains to be checked whether the partial parallelism between ocular and manual windows indexes a temporal coupling between different mechanisms involved in perception [43], but some conclusions can already be tentatively derived. Sensory alternations are supposed to trigger conscious alternations in perception [44] and the fact that there are faster alternations in the case of ocular temporal windows is consistent with the idea that sensory alternations are primarily driving perceptual switches. Top-down volitional control would have a role in gating these alternations by affecting their potential to reach consciousness. This interpretation, which is consistent with the current literature, implies that the ocular and manual windows are not independent but rather hierarchically organized. Caution is required, though, since ocular movements are not a pure measure of unconscious mechanisms, inasmuch they can be voluntarily controlled. Also, it remains to be investigated whether similar results, and especially the dissociation between ocular and manual temporal windows would be found with other types of static bistable figures.

It can be noted that the impact of the focus and switch conditions was significant, consistent with the literature [25,31,32] despite the short duration of the experiments. Such short experiments may thus be easily used in pathological groups to explore the rhythmic alternations in perception. Additionally eye tracking may be useful by providing a measure for no-report paradigms, in which no subjective report is required from the participants [6].

## Supporting information

**S1 File. Manual and ocular temporal windows dataset.**
(XLSX)

## Acknowledgments

We thank INSERM and the University Hospital of Strasbourg who provided support for this research.

## Author Contributions

**Conceptualization:** Luisa Weiner, Gilles Bertschy, Anne Giersch.

**Data curation:** Anne Giersch.

**Formal analysis:** Patrik Polgári, Jean-Baptiste Causin, Anne Giersch.

**Funding acquisition:** Anne Giersch.

**Investigation:** Jean-Baptiste Causin, Luisa Weiner.

**Methodology:** Anne Giersch.

**Project administration:** Anne Giersch.

**Resources:** Gilles Bertschy.

**Software:** Anne Giersch.

**Supervision:** Gilles Bertschy, Anne Giersch.

**Validation:** Patrik Polgári.

**Visualization:** Patrik Polgári.

**Writing – original draft:** Patrik Polgári.

**Writing – review & editing:** Patrik Polgári, Anne Giersch.

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
