## [Decision Letter · Decision Letter 0]

7 Nov 2019

PONE-D-19-24370

Novel method to measure temporal windows based on eye movements during viewing of the Necker cube

PLOS ONE

Dear Mr Polgári,

Thank you for submitting your manuscript to PLOS ONE. After careful consideration, we feel that it has merit but does not fully meet PLOS ONE’s publication criteria as it currently stands. Therefore, we invite you to submit a revised version of the manuscript that addresses the points raised during the review process.

We would appreciate receiving your revised manuscript by Dec 22 2019 11:59PM. To enhance the reproducibility of your results, we recommend that if applicable you deposit your laboratory protocols in protocols.io, where a protocol can be assigned its own identifier (DOI) such that it can be cited independently in the future. For instructions see: http://journals.plos.org/plosone/s/submission-guidelines#loc-laboratory-protocols

We look forward to receiving your revised manuscript.

Kind regards,

Marcello Costantini

Academic Editor

PLOS ONE

Journal Requirements:

Reviewers' comments:

Reviewer's Responses to Questions

**Comments to the Author**

1. Is the manuscript technically sound, and do the data support the conclusions?

Reviewer #1: No

Reviewer #2: Yes

2. Has the statistical analysis been performed appropriately and rigorously? 

Reviewer #1: Yes

Reviewer #2: Yes

3. Have the authors made all data underlying the findings in their manuscript fully available?

Reviewer #1: Yes

Reviewer #2: Yes

4. Is the manuscript presented in an intelligible fashion and written in standard English?

Reviewer #1: Yes

Reviewer #2: Yes

5. Review Comments to the Author

Reviewer #1: The topic of this paper is very interesting and matches current directions of research in the field. The manuscript is well written but in my opinion some issues should be adjusted in order to make it publicable.

- First sentence of the abstract: please change, it's quite raw and also wrong (alternate REGULARLY?)

- The new technique proposed could be very useful to boost No-Reports paradigms, but authors do not mention this in the manuscript, although they cite the paper of Tsuchiya in TICS 2015.

- The starting point of the authors (bistable perception with moving stimuli and eye movements) is not a firm point to start from, but rather a dangerous one!

- line 110 and 111: their > his/her

- Authors should try to explain why they did not found effects on vertical eye movements

- Similarly, they could speculate more and better on the faster oscillations of eye movements with respect to perception

- I'm skeptic about the first sentence of the conclusion: are you really convinced, after having carried out this piece of research, that eye movement can faithfully substitute explicit reports?

Reviewer #2: The study aims at providing implicit measures of the rate of perceptual reversals and reversal times during perception of bistable stimuli. To this aim, the authors asked twelve healthy subjects to give their manual responses while looking passively at a Necker cube, in four different conditions (spontaneous, focus, switch, control). During the experiment, they also recorded partipants' eye movements. Results suggest that recording of eye movements provide objective measures of time windows.

This is an interesting and well conducted study. I have only minor concerns, as reported below.

- In the introduction the authors claim that the definition of time windows is a critical issue to understand how conscious perception is structured in time. The authors propose that eye movents allow measuring unconscious perception of the Necker's cube. As the manual and ocular window do not correlate, would the authors conclude that the conscious and unconscious perception of bistable stimuli have different temporal structures?

- Is the interindividual variability of the "manual responses" comparable to the interindividual variability of the "eye responses"?

- What is the exact meaning of the "distance between the coordinates of the right and the left clusters in each condition"? How do the authors interpret this measure?

- How much do the authors believe their results about manual vs ocular temporal windows can be extended to bistable perception and perception in general?

- Please clarify the paragraph from line 392 to line 401. It is not clear to me how decisional mechanisms can be used to refrain from perceptual alternations and then manual responses.

6. PLOS authors have the option to publish the peer review history of their article (what does this mean?). If published, this will include your full peer review and any attached files.

Reviewer #1: No

Reviewer #2: No

---

## [Author Response · Author response to Decision Letter 0]

18 Dec 2019

Reviewer #1: The topic of this paper is very interesting and matches current directions of research in the field. The manuscript is well written but in my opinion some issues should be adjusted in order to make it publicable.

Our response: We would like to thank the reviewer for his/her positive comments, and for his/her suggestions, which helped us improve the paper.  

- First sentence of the abstract: please change, it's quite raw and also wrong (alternate REGULARLY?)

Our response: We agree with the reviewer and changed the first sentence of the abstract to (lines 31-32): ”Bistable stimuli can give rise to two different interpretations between which our perception will alternate.“ 

 

- The new technique proposed could be very useful to boost No-Reports paradigms, but authors do not mention this in the manuscript, although they cite the paper of Tsuchiya in TICS 2015.

Our response: We thank the reviewer for this thoughtful comment. We added the following at the end of the manuscript (lines 458-459): “Additionally eye tracking may be useful by providing a measure for no-report paradigms, in which no subjective report is required from the participants [6 (Tsuchiya et al., 2015)].”

 - The starting point of the authors (bistable perception with moving stimuli and eye movements) is not a firm point to start from, but rather a dangerous one!

Our response: Our starting point was to look for a complementary, objective measure to the subjective response of the subjects, we agree that the mechanisms underlying eye movements in the case of moving stimuli may be quite different. We tried to make this clearer in the Introduction (lines 70-75): “However, these studies relied on moving plaids or gratings and the recording of optokinetic nystagmus. Hence, it can be questioned whether ocular movements are entrained by the motion specifically. As a matter of fact, the tight coupling between perception and eye movements has been recently questioned [12 (Spering & Carrasco, 2015)], and dissociations between eye movements and conscious reports have been described even with moving stimuli [13 (Spering, Pomplun & Carrasco, 2011)].”

 - line 110 and 111: their > his/her

Our response: Edited as suggested.

 - Authors should try to explain why they did not found effects on vertical eye movements

Our response: This is an important point we forgot to clarify. In fact we made the analysis on y coordinates but chose not to report them. We added the following sentences in the Methods to justify our strategy (lines 241-245): “Y coordinates appeared to be redundant with x coordinates, hence we focused on ocular displacements on the horizontal axis. It is to be noted that an analysis performed on y coordinates led to similar but less clear results, probably because the two perceptual interpretations of the Necker cube differ more in their horizontal orientation (see similar observations in [26 (van Dam & van Ee, 2006)]).”

 

- Similarly, they could speculate more and better on the faster oscillations of eye movements with respect to perception

Our response: We agree with the reviewer and developed this point further in the Discussion when discussing the global difference between the number of eye movements and manual windows, we refer more to the literature (lines 389-392): “Regarding studies on bistable stimuli, our results are consistent with those reported by Brascamp et al. (2015) [36], or Kornmeier et al. (2019) [37] who showed in fMRI or EEG evidence of alternations that occur faster than the behavioral responses. As suggested by these authors sensory alternations may occur without reaching consciousness.”

We added a more speculative part towards the end of the Discussion (lines 407-417): “The literature suggests that a range of factors influence alternations and perception in general [40]. For example, it has been demonstrated that top-down factors, or volitional control [7,25,27,31,32,41,42] can modify the rate of perceptual alternations, as also observed in our focus and switch conditions. As repeatedly observed in focus conditions [43], it is however impossible to completely suppress alternations, confirming the conjoint influences of sensory competition and volitional control. In our study one possible explanation may thus be that the ocular temporal windows reflect an irrepressible sensory alternation, whereas manual responses would be more susceptible to volitional control. The latter may be used to refrain from perceptual alternations, by vetoing sensory influences, or may be necessary to transform a sensory alternation into a conscious one. Both explanations would account for the ocular fixation clusters’ lack of sensitivity to the focus condition, contrasting with the increase in length for the manual time windows.”

 

- I'm skeptic about the first sentence of the conclusion: are you really convinced, after having carried out this piece of research, that eye movement can faithfully substitute explicit reports?

Our response: The reviewer is right to point out this sentence and we believe the text was not clear enough on this point. We added the following precisions (lines 433-437): “We do not claim that eye movements analyzed the way we did could substitute explicit reports, but instead we believe that eye movements could serve as a complementary measure to explicit reports, inasmuch our ocular temporal windows (ocular clusters) seem to reveal dynamic perceptual mechanisms at a different, more automatic level than the one of conscious perception and perceptual alternations.”

Reviewer #2: The study aims at providing implicit measures of the rate of perceptual reversals and reversal times during perception of bistable stimuli. To this aim, the authors asked twelve healthy subjects to give their manual responses while looking passively at a Necker cube, in four different conditions (spontaneous, focus, switch, control). During the experiment, they also recorded partipants' eye movements. Results suggest that recording of eye movements provide objective measures of time windows.  This is an interesting and well conducted study. I have only minor concerns, as reported below.

Our response: We thank the reviewer for his/her positive as well as helpful comments.  

- In the introduction the authors claim that the definition of time windows is a critical issue to understand how conscious perception is structured in time. The authors propose that eye movents allow measuring unconscious perception of the Necker's cube. As the manual and ocular window do not correlate, would the authors conclude that the conscious and unconscious perception of bistable stimuli have different temporal structures?

Our response: Thank you for this thoughtful comment. We have to remain cautious, since the response of the participants include both unconscious and conscious perceptual aspects, as well as intentions, e.g. voluntary decisions based on ‘focus’ or ‘switch’ task instructions. Ocular movements, on the other hand, are not purely unconscious. We thus cannot equate the manual responses with conscious perception, and ocular clusters with unconscious perception. We thus tried to speculate cautiously (lines 445-452): “Sensory alternations are supposed to trigger conscious alternations in perception [45] and the fact that there are faster alternations in the case of ocular temporal windows is consistent with the idea that sensory alternations are primarily driving perceptual switches. Top-down volitional control would have a role in gating these alternations by affecting their potential to reach consciousness. This interpretation, which is consistent with the current literature, implies that the ocular and manual windows are not independent but rather hierarchically organized. Caution is required, though, since ocular movements are not a pure measure of unconscious mechanisms, inasmuch they can be voluntarily controlled.”

 - Is the interindividual variability of the "manual responses" comparable to the interindividual variability of the "eye responses”?

Our response: Thank you for the pertinent comment. We performed an ANOVA on the Standard Deviation of the duration for manual and ocular windows and did not find any significant main effect of the type of window (manual or ocular), or an interaction with the experimental conditions, suggesting similar interindividual variability for the two measures.

 - What is the exact meaning of the "distance between the coordinates of the right and the left clusters in each condition"? How do the authors interpret this measure?

Our response: We agree that our wording may have been ambiguous. The distance refers to the spatial separation between the two ocular clusters, and is operationalized through the difference between the coordinates of the two ocular clusters, i.e. groups of successive fixations on the right and left side of the Necker Cube (see modified Fig 2). We tried to clarify this point in both the Methods and Results sections:

(lines 259-261): “In the other three conditions, we calculated the median x coordinates of the left and right fixation clusters, and the spatial distance between the left and right fixation clusters (median x coordinate for the right cluster – median x coordinate for the left cluster, see Fig 2).”

(lines 297-299): “To understand the origin of the interaction, we calculated the distance between the right and left clusters in each condition (median x coordinates for the right cluster –median x coordinates for the left cluster) (Table 1, also illustrated in Fig 2).”

This measure informs us on the perceptual separation associated with each interpretation of the Necker Cube. The fact that this separation is similar in the control condition and in the bistable condition validates the link between the percept and the ocular fixations.

 

- How much do the authors believe their results about manual vs ocular temporal windows can be extended to bistable perception and perception in general?

Our response: Inasmuch our results suggest that sensory alternations need additional control to cross the consciousness threshold, they are consistent with a broad literature. Whether ocular measures can be applied the same way to other bistable stimuli remains to be investigated, at least for static stimuli. We mention this point in the conclusion (lines 452-454): “Also, it remains to be investigated whether similar results, and especially the dissociation between ocular and manual temporal windows would be found with other types of static bistable figures.”

 - Please clarify the paragraph from line 392 to line 401. It is not clear to me how decisional mechanisms can be used to refrain from perceptual alternations and then manual responses.

Our response: Thank you for pointing out this section which was not clear enough. We agree that ‘decisional mechanisms’ was a poor choice of words, and we reformulated this section (lines 407-417): “The literature suggests that a range of factors influence alternations and perception in general [40]. For example, it has been demonstrated that top-down factors, or volitional control [7,25,27,31,32,41,42] can modify the rate of perceptual alternations, as also observed in our focus and switch conditions. As repeatedly observed in focus conditions [43], it is however impossible to completely suppress alternations, confirming the conjoint influences of sensory competition and volitional control. In our study one possible explanation may thus be that the ocular temporal windows reflect an irrepressible sensory alternation, whereas manual responses would be more susceptible to volitional control. The latter may be used to refrain from perceptual alternations, by vetoing sensory influences, or may be necessary to transform a sensory alternation into a conscious one. Both explanations would account for the ocular fixation clusters’ lack of sensitivity to the focus condition, contrasting with the increase in length for the manual time windows.”

---

## [Editor Report · Decision Letter 1]

20 Dec 2019

Novel method to measure temporal windows based on eye movements during viewing of the Necker cube

PONE-D-19-24370R1

Dear Dr. Polgári,

We are pleased to inform you that your manuscript has been judged scientifically suitable for publication and will be formally accepted for publication once it complies with all outstanding technical requirements.

With kind regards,

Marcello Costantini

Academic Editor

PLOS ONE

---

## [Editor Report · Acceptance letter]

27 Dec 2019

PONE-D-19-24370R1 

Novel method to measure temporal windows based on eye movements during viewing of the Necker cube 

Dear Dr. Polgári:

I am pleased to inform you that your manuscript has been deemed suitable for publication in PLOS ONE. Congratulations! Your manuscript is now with our production department. 

With kind regards,

on behalf of

Dr. Marcello Costantini 

Academic Editor

PLOS ONE